# Advance care planning in multiple sclerosis (ConCure-SM): A multicenter single-arm pilot and feasibility study

Alessandra Solari[1]*, Ludovica De Panfilis[2,3], Roberta Martina Zagarella[4], Luca Ghirotto[5], Mariangela Farinotti[1], Alberto Gajofatto[6,7], Maria Grazia Grasso[8], Paola Kruger[9], Alessandra Lugaresi[10,11], Katia Mattarozzi[12], Sara Montepietra[13], Francesco Patti[14], Eugenio Pucci[15], Michela Rimondini[16], Claudio Solaro[17], Marta Perin[18], Andrea Giordano[1,19○], Simone Veronese[20○], on behalf of the ConCure-SM project[¶]

1 Unit of Neuroepidemiology, Fondazione IRCCS Istituto Neurologico Carlo Besta, Milan, Italy, 2 Department of Medical and Surgical Sciences, University of Bologna, Bologna, Italy, 3 Azienda Ospedaliero-Universitaria IRCCS Bologna, Bologna, Italy, 4 National Research Council (CNR), Interdepartmental Center for Research Ethics and Integrity (CID Ethics), Rome, Italy, 5 Qualitative Research Unit, Azienda USL-IRCCS di Reggio Emilia, Reggio Emilia, Italy, 6 Department of Neuroscience, Biomedicine and Movement Sciences, University of Verona, Verona, Italy, 7 Unit of Neurology, Borgo Roma Hospital, Azienda Ospedaliera Universitaria Integrata Verona, Verona, Italy, 8 Multiple Sclerosis Unit, IRCCS S. Lucia Foundation, Rome, Italy, 9 The European Patients' Academy (EUPATI), Rome, Italy, 10 IRCCS Istituto delle Scienze Neurologiche di Bologna, Bologna, Italy, 11 Dipartimento di Scienze Biomediche e Neuromotorie, Università di Bologna, Bologna, Italy, 12 Department of Medical and Surgical Sciences, Alma Mater Studiorum University of Bologna, Bologna, Italy, 13 Multiple Sclerosis Center, Azienda USL-IRCCS di Reggio Emilia, Reggio Emilia, Italy, 14 University Hospital Policlinico Vittorio Emanuele, Catania, Italy, 15 UOC Neurologia, AST Fermo, Fermo, Italy, 16 Section of Clinical Psychology, Department of Neuroscience, Biomedicine and Movement Sciences, University of Verona, Policlinico G.B. Rossi, Verona, Italy, 17 Neurology Unit, Galliera Hospital, Genoa, Italy, 18 Bioethics Unit, Azienda USL-IRCCS di Reggio Emilia, Reggio Emilia, Italy, 19 Neurology, Public Health and Disability Unit, Fondazione IRCCS Istituto Neurologico Carlo Besta, Milan, Italy, 20 Fondazione FARO ETS, Turin, Italy

○ These authors contributed equally to this work.
¶ Membership of the ConCure-SM project is provided in the Acknowledgements.
* alessandra.solari25@gmail.com

## Abstract

### Background

Advance care planning (ACP) practice in people with progressive multiple sclerosis (PwPMS) remains limited. ConCure-SM project aims to assess the effectiveness of a structured ACP intervention (clinician's training programme and use of a booklet during ACP conversations) using a multi-phased design.

### Methods

Single-arm pilot/feasibility trial involving PwPMS, their significant others (SOs), and clinicians from six Italian centers. Primary study outcome was completion of an advance care plan document (ACP-Doc). Other outcomes included safety, feasibility of enrollment and assessment, and (analyzed using mixed-methods approach)

**Data availability statement:** All relevant data underlying the findings of this study are available in the Zenodo repository at https://doi.org/10.5281/zenodo.17144481, and in the Supporting information files.

**Funding:** The study was supported by FISM - Fondazione Italiana Sclerosi Multipla – cod. 2020/R-Multi/024 and financed or co-financed with the '5 per mille' public funding to AS. The funding source had no role in study design, data collection, data analysis, data interpretation or report writing. We also thank the Italian Ministry of Health (RRC). There was no additional external funding received for this study.

**Competing interests:** AL reports grants from Novartis, during the conduct of the study; personal fees from Biogen, Merck Serono, Mylan, Novartis, Roche, Sanofi/Genzyme, Teva and FISM. FP received personal compensation for serving on advisory board and/or speaking activities by Almirall, Bayer, Biogen, Bristol Meyers & Squibb, Merck, Novartis Roche, Sanofi and TEVA; he further received research grants by Biogen Italy, Biogen Global, Merck, University of Catania, FISM and Reload Onlus Patients Association. AS received personal compensation for serving on advisory board and speaking activities by Almirall and Merck. All the other authors have no competing interests. This does not alter our adherence to PLOS ONE policies on sharing data and materials.

Hospital Anxiety and Depression Scale (HADS), quality of communication, quality of life (MSQOL-29), and caregiver burden. Participants were interviewed to identify factors influencing the ACP implementation process.

## Results

Seventy-five PwPMS were eligible out of 164 screened; 56/75 (75%) refused participation and 19 were included. Of these, 11 (58% vs 30% hypothesized) completed the ACP-Doc. A total of 25 adverse events (increase in anxiety) occurred, three possibly related to the intervention, and we found a worsening of HADS-Anxiety score ($p=0.02$) and MSQOL-29 mental health composite score ($p=0.04$) during follow-up. PwPMS/SO interviews revealed four themes: significance of the ACP process (on the individual, on relation with clinicians), its impact (on emotions, on family relations), preparedness as key, and challenges (practicability, SO commitment). Barriers and facilitators for ACP were identified in two clinician focus groups.

## Conclusions

The intervention supported neurologists in guiding PwPMS in their ACP. However, trial findings and the high proportion of refusals point to the need to enrich the intervention with a new component targeting PwPMS and SOs.

## Trial registration

ISRCTN48527663.

## Introduction

Multiple sclerosis (MS) is the most common cause of progressive neurological disability in young adults [1]. Around 15% of MS sufferers have a primary progressive course at diagnosis, and a further 35% develop secondary progressive disease after 15 years [2]. People with progressive form (PwPMS) may live for many years experiencing a wide range of symptoms, impairments (including cognitive impairment) and comorbidities [3,4] and may benefit from advance care planning (ACP).

ACP is a process that "enables individuals who have decisional capacity to identify their values, to reflect upon the meanings and consequences of serious illness scenarios, to define goals and preferences for future medical treatment and care, and to discuss these with family and healthcare professionals" [5]. Consistently with the shared decision-making model [6], ACP involves both the patient and their clinician. Together, they make informed decisions about the patient's (future) care. If the patient wishes, their significant other (SO) can also be involved. ACP differs from general medical decision-making in that it is based on an anticipated health deterioration, and includes a focus on the person's wishes and preferences for the time when they may lose decisional capacity. The planning process helps the patient to identify their own personal values and goals, understand their health status, and the available treatment/

care options. Finally, ACP encourages discussion around end of life (EOL) care – a subject that is generally not considered part of health care planning, and one that can be avoided by both patients and clinicians. The ACP process may result in the patient choosing to produce an advance care plan document (ACP-Doc) and to appoint a trustee (or else).

ConCure-SM is a project aimed to set up and evaluate the efficacy of a structured ACP intervention for PwPMS consisting of a training programme for MS clinicians and a booklet to be used during the ACP conversations. The theoretical basis of the project is the shared decision-making model [6], and its methodological basis is the Medical Research Council framework for developing and evaluating complex interventions [7]. In a previous project phase, we co-developed the ConCure-SM booklet [8] and translated-adapted two self-reported outcome measures (PROMs) not available in Italian [9]. In this pilot phase we assessed the feasibility, acceptability, and preliminary effectiveness of the intervention hypothesizing that it would support completion of an ACP-Doc, increase congruence in treatment preferences between PwPMS and their carers, and the quality of communication. The objectives of the pilot phase are reported in the Box 1.

## Materials and methods

### Study design and setting

Between March 8, 2022 and March 6, 2023, we conducted a multicenter single-arm explanatory sequential mixed methods pilot and feasibility trial. The study protocol was approved by the ethics committees of the coordinating center, Fondazione IRCCS Istituto Neurologico Carlo Besta (internal ref. 83/2021), and all the six enrolling centers: Verona (55917), Moncrivello (15210), Reggio Emilia (80829), Bologna (90076), Rome (921), Catania (47839). The enrolling centers are located in northern (four centers), central and southern Italy (one center each). Two centers are rehabilitation hospitals (one of which a research hospital), three are MS centers (two university hospitals, one research hospital) and one is a rehabilitation and MS center from a research hospital. The study followed the CONSORT guidance (S1 File) [10,11], and was carried out in accordance with the Good Clinical Practice principles and the Declaration of Helsinki recommendations. All participants (PwPMS, SOs, neurologists, and other clinicians) gave written informed consent.

### Intervention

The goal of the intervention was to prime clinicians to discuss goals of care and ACP with PwPMS. It consisted of a clinician training programme and use of the ConCure-SM booklet [8]. The Italian Law 219/217 (Article 5) prescribes that ACP involves the patient, their referring physician, and (when applicable) the trustee. To promote ACP knowledge within the MS team, we also trained non-physician healthcare professionals. Trainers were researchers, clinicians and bioethicists experienced in leading courses and workshops on patient-clinician communication and ACP.

The training programme consisted of three components: 1) A residential, Continuing Medical Education accredited course of one-and-half days (12 hours) duration. It included: one 2.5-hour theoretical session on the clinical, ethical and statutory principles of shared decision-making and ACP; two 4-hour empirical sessions (one on each day) on conducting ACP conversations in various clinical scenarios using the ConCure-SM booklet through guided role play exercises; two 45-minute self-evaluation sessions (at the beginning and at the end of the programme). 2) Four booster sessions (teleconference) followed the residential training, each lasting 90–120 minutes. Trainees shared their experiences and discussed difficult cases, guided by two trainers. 3) One-to-one videoconference or telephone calls with trainers for issues emerging during ACP conversations.

The ConCure-SM booklet [8] consists of an introduction, a 'guidance', and the ACP-Doc, which the PwPMS and their referring physician must complete electronically or manually.

### Assessments and participants

There was a baseline assessment, a first ACP conversation taking place within one month from baseline, and a follow-up assessment within one week of the first ACP conversation (T1) and six months thereafter (T2). The baseline and follow-up

assessments were performed via a web-based trial platform [9] that contained the case report form (eCRF) and the PROMs. The physician recorded on the platform subsequent ACP conversations that occurred during follow-up.

**Eligibility and screening.** Adult PwPMS were included if they were able to communicate in Italian, had one or more of seven conditions that would make ACP relevant (S1 Box), and gave written consent [9]. PwPMS were excluded if they had severe cognitive compromise (MMSE < 19), impairments preventing communication, psychosis or other serious psychiatric conditions, or if they had already completed an ACP-Doc. The PwPMS could involve their SO (family member, relative, or friend, who is next of kin or is key decision-maker as designated by the PwPMS).

Before baseline assessment, PwPMS and SOs gave signed informed consent. Then they completed the baseline questionnaires/instruments by accessing the trial platform or via a telephone call with one trained interviewer.

Each center collected information on the number of PwPMS and SOs approached, screened, and eligible prior to enrollment, with reasons for non-enrolment.

**The ACP conversation.** The first ACP conversation was scheduled at the center, was intended to be one hour long, and was audio-recorded. It involved the PwPMS, the ACP-trained neurologist, and, when applicable, the SO. In addition, another ACP-trained clinician participated, if the PwPMS agreed. Subsequent conversations were documented/noted in the eCRF.

**Outcome measures.** A range of measures were collected to capture the full process of ACP and whether the ConCure-SM intervention had any effect on completion of an ACP-Doc (primary outcome measure), quality of patient-clinician communication, and caregiver burden. In addition, as a study-related increase in emotional burden could not be excluded, serious adverse events (SAE: admission to psychiatric ward, suicide attempt, death) were monitored by an independent Data and Safety Monitoring Committee.

We used the published Italian version of the following inventories: Control Preference Scale (CPS) [12,13]; Hospital Anxiety and Depression Scale (HADS) [14,15]; Observing Patient Involvement in Decision Making (OPTION) [16,17]; 29-item Multiple Sclerosis Quality of Life (MSQOL-29) [18]; Zarit Burden Interview (ZBI) [19,20]. The 4-item ACP-Engagement and the Quality Of Communication (QOC) inventories were translated/culturally adapted from source language [21–23].

We assessed the quality of the first ACP conversation (which was unobtrusively audio-recorded and transcribed verbatim) considering three perspectives: an independent observer, the PwPMS, and the physician (manuscript in preparation).

**Nested qualitative study.** We conducted one-on-one semi-structured interviews with PwPMS and SOs, and two focus groups of clinicians involved in intervention delivery. Interviews and focus groups were held via videoconference to ease participation of PwPMS with high disability and SOs with caregiving commitments, as well as clinicians. PwPMS and/or SOs who had difficulty in using personal computer or other devices, were interviewed on the telephone. All participants were informed of the aim and procedure of the interview/focus group, and provided written consent. Further details are reported in the (published [9]) protocol (S2 File), in S3 (Interview/ focus group guides) and S4 Files (COREQ checklist) [24].

## Data analysis

**Study power.** There are no data available on the occurrence of ACP in MS: by hypothesizing a proportion in the PwPMS population of 10%, a sample size of 35 subjects achieves a power of 90%, assuming a type I error of 5%, to detect a proportion of ACP-Doc of 30%. By hypothesizing a proportion in the PwPMS population of 8%, a sample size of 35 subjects achieves a power of 95%, assuming a type I error of 5%, to detect a proportion of ACP-Doc of 30%. Considering drop-outs and withdrawals, we aimed to recruit at least 40 PwPMS.

**Statistics.** Categorical variables were compared using the chi-squared or Fisher's exact test. Within-group comparisons were carried out using the Wilcoxon signed-rank test.

Our primary end-point was the proportion of PwPMS completing an ACP-Doc during the six-month period. Change in the secondary outcome measures were calculated using generalized estimating equations. Data were analyzed according

to the intention-to-treat principle, using multiple imputation of missing values. A *p*-value < 0.05 was considered statistically significant. No correction for multiple comparisons was applied. All analyses were performed using STATA 16 (College Station, Texas 77845 USA).

**Qualitative data.** Interviews and focus groups were audio-recorded and transcribed verbatim. Five researchers (MP, RMZ, SV, LDP, and LG) analyzed interviews and focus groups using thematic analysis [25], with interpretation guided by the Normalization Process Theory (NPT) [26].

**Qualitative analysis.** The data material was analyzed line-by-line using inductive coding to identify factors influencing the process of ACP. After that, the factors were mapped to the constructs of the NPT and their specific domains as a conceptual framework that explains implementation processes [27]. NPT identifies four essential determinants of 'normalizing' complex interventions into clinical practice: *coherence* (the extent to which an intervention is understood as being meaningful, achievable and valuable); *cognitive participation* (the engagement of clinicians necessary to deliver the intervention); *collective action* (the work that brings the intervention into use); and *reflexive monitoring* (the ongoing process of adjusting the intervention to keep it in place) [26]. Before the focus groups, clinicians completed the Normalisation MeAsure Development (NoMAD) questionnaire [28]. The NoMAD consists of 23 items assessing the four NPT domains from the perspective of professionals directly involved in the work of implementing complex healthcare interventions.

## Results

### Enrolment and participants' characteristics at baseline

164 PwPMS were screened, and 89 were excluded. Of 75 eligible patients, 19 (25%) accepted to participate and were analyzed (Fig 1).

Details of the screened PwPMS across centers are reported in S1 Table. Characteristics of enrolled PwPMS at baseline, SOs, ACP-trained neurologists, and centers are reported in Table 1. Participants' characteristics by center are reported in S2 Table. CPS data were not available due to a technical problem.

### The ACP conversations and primary outcome

Twenty-four ACP conversations occurred during follow-up: 19 first ACP conversations, and 5 second conversations. Median time from baseline to first conversation was 20 days, from the first to the second conversation 27 days. The main features of the conversations are reported in Table 2.

Overall, 11/19 (58%) PwPMS completed the ACP-Doc over the six-month period: seven during the first and four during the second ACP conversation.

Mean QOC subscale scores (0–100) were 92.4 (SD 9.6) for 'general communication' and 90.1 (SD 10.2) for 'communication about EOL care'. Mean QOC-SO scores were 95.2 (SD 6.4) for 'general communication' and 94.3 (SD 7.0) for 'communication about EOL care'. Finally, mean QOC-Doc score was 69.5 (SD 16.0) (Table 2).

### Adverse events and attrition

There was one SAE (hospitalization for pneumonia and pre-coma), unrelated to the intervention. There were 25 adverse events (Table 3) in 13 PwPMS; 3/25 (increase in anxiety) were deemed to be possibly related to the intervention.

HADS-Anxiety score increased significantly over time (p = 0.02), while this was not the case for HADS-Depression score (p = 0.46) (Table 3; S1 and S2 Figs).

One patient refused conversation audio-recording and did not complete PROMs. One patient did not complete PROMs at 6-month follow-up. Two SOs did not complete ZBI at baseline. Two SOs did not complete ZBI at 6-month follow-up (Fig 1).

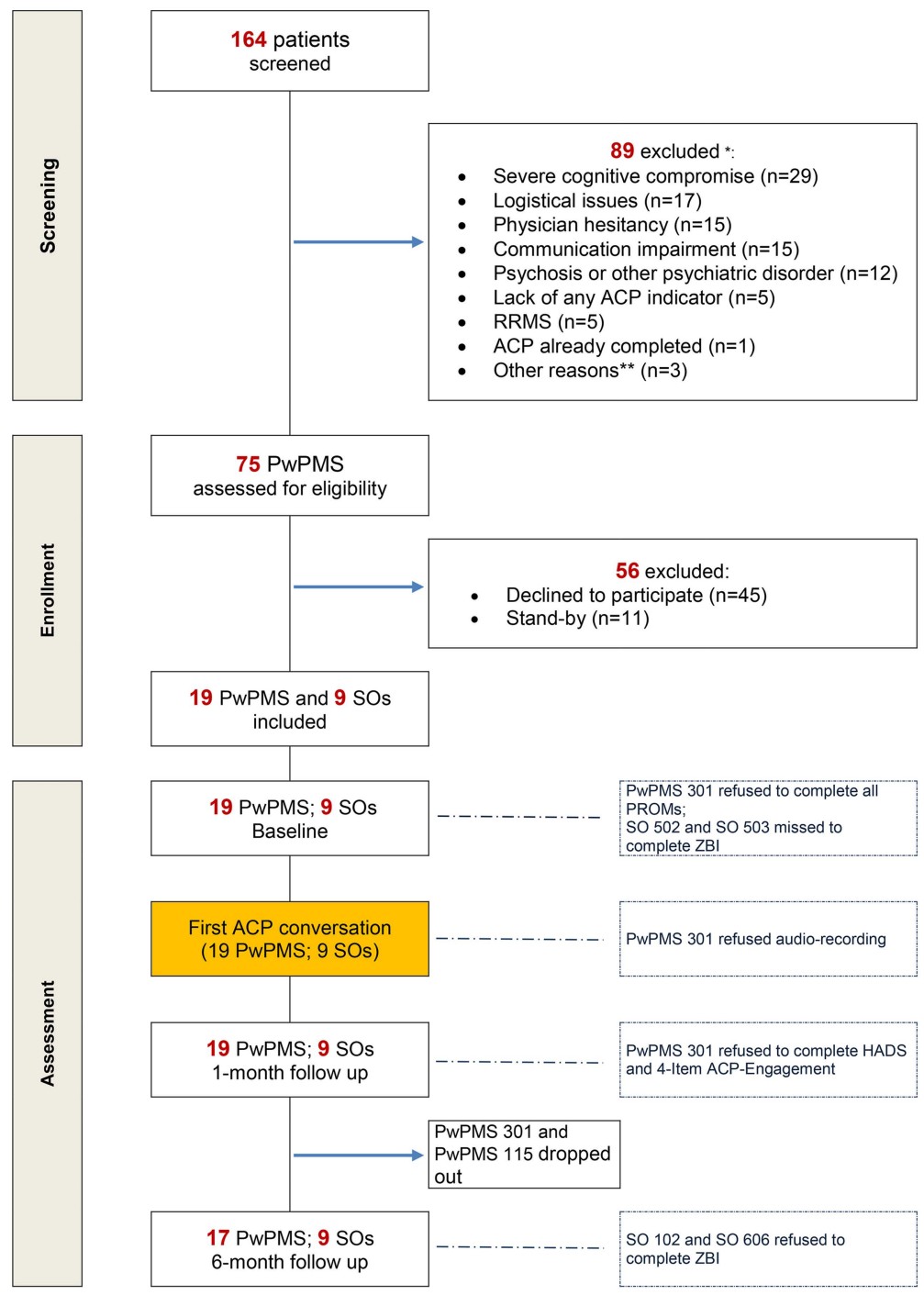

**Fig 1. CONSORT 2010-SPI flow diagram of the trial.** ACP, advance care planning; PROM, patient-reported outcome; RRMS, relapsing remitting multiple sclerosis; SO, significant other; ZBI, Zarit Burden Interview. Boxes with dotted line identify missing PROMs.

**Table 1. Baseline characteristics of the persons with progressive multiple sclerosis (PwPMS), significant others (SOs), the programme-trained caring physicians, and centers.**

| Characteristic | PwPMS (n=19) | SOs (n=9) | | Physicians (n=7) |
|---|---|---|---|---|
| Women | 8 (42%) | 4 (44%) | | 5 (71%) |
| Age (years) – Mean, SD | 61.6 (7.6) | 57.1 (11.5) | | 41.7 (12.3) |
| Education: Primary (5–8 years) | 2 (10%) | 0 (0%) | MS experience (years)- Median (IQR) | 10 (5-26) |
| Secondary (12–13 years) | 14 (74%) | 6 (67%) | | |
| College/university (14 + years) | 3 (16%) | 3 (33%) | | |
| Occupation: Retired (disability) | 11 (58%) | 0 (0%) | | |
| Employed | 4 (21%) | 5 (56%) | | |
| Retired (age) | 3 (16%) | 2 (22%) | | |
| Housewife | 1 (5%) | 2 (22%) | | |
| Relation: Spouse/partner | | 6 (67%) | | |
| Other relative | | 2 (22%) | | |
| Son | | 1 (11%) | | |
| 22-item ZBI – Median (IQR) | | 24 (10-34) | | |
| Age at MS diagnosis (years) – Mean, SD | 40.3 (14.4) | | | |
| MS type: Primary progressive | 8 (42%) | | | |
| Secondary progressive | 11 (58%) | | | |
| EDSS – Median (IQR) | 8.0 (6.5-8.0) | | | |
| Barthel Index – Median (IQR) | 29.0 (14.0-61.5) | | | |
| HADS Anxiety – Median (IQR) | 4.0 (2.0-5.0) | | | |
| Depression – Median (IQR) | 4.5 (1.0-8.0) | | | |
| MSQOL-29 Physical Health Composite – Median (IQR) | 31.0 (36.7-40.3) | | | |
| Mental Health Composite – Median (IQR) | 60.9 (49.3-68.3) | | | |
| **Centers (n=6)** | | | | |
| MS rehabilitation center | 3 (50%) | | | |
| MS clinical center | 2 (33%) | | | |
| Rehabilitation center | 1 (17%) | | | |
| PwMS followed – Median (IQR) | 911 (350-2070) | | | |
| PwPMS followed – Median (IQR) | 375 (218-1000) | | | |

EDSS, Expanded Disability Status Scale; HADS, Hospital Anxiety and Depression Scale; IQR, interquartile range; MSQOL-29, Multiple Sclerosis Quality of Life-29 items; MS, multiple sclerosis; SD, standard deviation; ZBI, Zarit Burden Interview.

## Other outcome measures

There were no differences for 4-item ACP Engagement survey scores over time ($p=0.47$) (Table 3; S3 Fig), and for MSQOL-29 physical health composite scores ($p=0.32$), whereas there was a significant worsening for MSQOL-29 mental health composite scores over time ($p=0.04$) (Table 3; S4 Fig). There were no differences for ZBI scores over time ($p=0.44$) (Table 3; S5 Fig). These findings should be interpreted with caution (see Discussion, 'Study limitations' section) due to small sample size and multiplicity problem.

Per-protocol analysis findings matched those of the main analysis for all outcomes (S3 Table).

## Qualitative findings and process evaluation

Five PwPMS and four SOs were interviewed (interviewees' characteristics are not given to prevent their identification); two were dyads. Interview guides are reported in S3 File. Representative quotations are shown in Table 4.

**Table 2. Main features of the advance care planning (ACP) conversations overall and across centers.**

| First conversation | Overall (n = 19) | Verona (n = 2) | Moncrivello (n = 3) | Reggio Emilia (n = 3) | Bologna (n = 2) | Rome (n = 5) | Catania (n = 3) |
|---|---|---|---|---|---|---|---|
| | N (%) | | | | | | |
| Time from inclusion (days)[†] | 20 (15.5-31) | 20 (11-45.5) | 45 (36.5-47) | 15 (14.5-18) | 16 (16−16) | 16 (14-17) | 30 (29-35.5) |
| Duration (min)[1] | 62.5 (45-100) | 80.5 (76-85) | 60 (56-60) | 65 (65-90) | 87.50 (75-100) | 60 (60−60) | 75 (45-75) |
| Significant other involved | 9 (50) | 1 (50) | 0 | 0 | 1 (50) | 5 (100) | 2 (67) |
| Another clinician involved | 9 (50) | 1 (50) | 0 | 2 (67) | 1 (50) | 2 (40) | 3 (100) |
| Interruptions, n | 3 (17) | 0 | 0 | 3 (100) | 0 | 0 | 0 |
| End of life discussed | 14 (78) | 2 (100) | 3 (100) | 3 (100) | 2 (100) | 3 (60) | 1 (33) |
| Outcome of conversation | | | | | | | |
| ACP completed | 7 (39) | 1 (50) | 0 | 3 (100) | 2 (100) | 1 (20) | 0 |
| Need another conversation | 11 (61) | 1 (50) | 3 (100) | 0 | 0 | 4 (80) | 3 (100) |
| New conversation scheduled | 2 (13) | 1 (50) | 1 (33) | 0 | 0 | 0 | 0 |
| QOC-Doc 1 *'How comfortable do you feel when talking about dying?'*[‡] | 6.9 (2.1) | 7.3 (0.6) | 8.3 (0.6) | 8.0 (0) | 8.5 (0.7) | 6.2 (2.5) | 4.0 (1.7) |
| QOC-Doc 2 *'Overall, how would you rate your communication with this patient during the ACP conversation?'*[2] | 7.0 (1.2) | 7.3 (0.6) | 7.7 (0.6) | 7.3 (0.6) | 8.0 (0) | 6.2 (2.2) | 6.3 (0.6) |
| QOC-Doc total score (0–100) | 69.5 (16.0) | 73.3 (5.8) | 80.0 (5.0) | 76.7 (2.9) | 82.5 (3.5) | 62.0 (23.1) | 51.7 (10.4) |
| **Second conversation** | Overall (n = 5) | Verona (n = 1) | Moncrivello (n = 3) | | | | Catania (n = 1) |
| Time from first conversation (days)[1] | 27 (24.2-40.2) | 22 | 27 (26-28) | – | – | – | 74 |
| Duration (min)[1] | 50 (45-60) | 60 | 50 (47.5-55) | – | – | – | 30 |
| Significant other involved | 1 (25) | | | | | | |
| Another clinician involved | 0 | – | – | – | – | – | – |
| Interruptions, n | 0 | – | – | – | – | – | – |
| Outcome of conversation | | | | | | | |
| ACP-Doc completed | 4 (80) | 1 (100) | 3 (100) | – | – | – | 0 |
| New conversation scheduled | 1 (20) | 0 | 0 | – | – | – | 1 (100) |

QOC, quality of communication questionnaire.

[†] Median (interquartile range).

[‡] Mean (standard deviation).

All PwPMS and SOs recognized the existential significance of the ACP process. The ACP conversation strengthened the partnership with the caring clinicians, and participants felt supported by them during enrollment and discussions, appreciating clinicians' clarity and competence. However, engaging in ACP came with emotional costs. Eight out of nine interviewees experienced the discussion of EOL topics as demanding despite appreciating its value. One SO (and trustee) felt embarrassed, likening his role to a "squire" protecting his loved one from strong emotions. Another SO (and trustee) worried about his future "emotional resilience" and described his role as "heavy". Generally, ACP discussion and family relationships influenced each other. Specifically, in some instances, ACP facilitated broader family conversations about EOL decisions, and in other instances, discussion within the family impacted PwPMS choices. Finally, most of the PwPMS felt supported by involving the family in the ACP discussion, and SOs felt relieved by beginning such discussion. Preparedness (of PwPMS and SOs) reduced anxiety during discussions, while lack of preparedness increased difficulty and led to negative emotions. Preparedness (of PwPMS and SOs) reduced anxiety during discussions, while lack of preparedness increased difficulty and led to negative emotions. In one case, misunderstanding about PwPMS preparedness

**Table 3. Secondary outcome measures and safety data.**

| Secondary outcome measures | Baseline | | First ACP conversation | | 6-month follow up | | P value |
|---|---|---|---|---|---|---|---|
| | *n* | *Mean, median (IQR)* | *n* | *Mean, median (IQR)* | *n* | *Mean, median (IQR)* | |
| **Persons with progressive multiple sclerosis**† | | | | | | | |
| HADS-Anxiety | 19 | 4.9, 4 (2-7) | 19 | 5.6, 6 (3-8) | 19 | 7.1, 6 (3-10) | **0.02** |
| HADS-Depression | 19 | 5.6, 4 (1-8) | 19 | 6.3, 6 (2-9) | 19 | 6.5, 7 (3.4-8) | 0.46 |
| 4-item ACP-Engagement | 19 | 12, 13 (7-16) | 19 | 13.4, 13 (9-20) | 19 | 13.5, 13 (9-20) | 0.47 |
| MSQOL-29, PHC | 19 | 37, 37.5 (33-42.5) | – | – | 19 | 33.8, 35.5 (24.5-40) | 0.32 |
| MSQOL-29, MHC | 19 | 59, 59.5 (48.0-68.4) | – | – | 19 | 50.6, 52.3 (43.4-64.3) | **0.04** |
| **Significant others**‡ | | | | | | | |
| ZBI total score | 9 | 23.9, 24 (13-34) | 9 | 23, 23 (13-30) | 9 | 20.7, 19 (14-27) | 0.44 |

**Safety data**

| Center | Adverse events | | PwPMS | Relatedness | | | |
|---|---|---|---|---|---|---|---|
| | Not serious | Serious | | Possibly/Probably related | Not related | Unsure | |
| Verona | 4 | 0 | 2/3 | 0 | 4 | 0 | |
| Moncrivello | 4 | 0 | 2/3 | 0 | 4 | 0 | |
| Reggio Emilia | 1 | 0 | 1/3 | 0 | 0 | 1 | |
| Bologna | 3 | 1§ | 2/2 | 1 | 3 | 0 | |
| Rome | 7 | 0 | 3/5 | 0 | 7 | 0 | |
| Catania | 5 | 0 | 3/3 | 1 | 4 | 0 | |
| Totals | 24 | 1§ | 13/19 | 2 | 22 | 1 | |

Significant values are reported in bold.

ACP, advance care planning; HADS, Hospital Anxiety and Depression Scale; IQR, interquartile range; MHC, mental health composite; MSQOL-29, 29-item Multiple Sclerosis Quality of Life; PHC, physical health composite; ZBI, Zarit Burden Interview.

†Generalized estimating equations; missing values imputed using auxiliary variables age, time visit, gender, education, disability level, and disease duration.

‡Generalized estimating equation; missing values imputed using auxiliary variables significant other's age, gender, education, time visit, PwPMS age, disability level, and disease duration.

§Serious adverse event: Intensive care hospitalization for pneumonia and pre-coma (unrelated to the intervention).

Description of the adverse events (AEs):

• 13 AEs were increase >20% in the HADS Anxiety score between baseline and 1-month follow up or between 1- and 6-month follow ups; 2 AEs were possibly/probably related, 11 unrelated; in 5 AEs the score was >8 at post-test.

• 10 AEs were increase >20% in the HADS Depression score between baseline and 1-month follow up or between 1- and 6-month follow ups; all AEs were unrelated; in 3 AEs the score was >8 at post-test.

• 1 AE was a feeling of anxiety and distress communicated to the neurologist in a follow-up visit. It was judged as possibly related to the first advance care planning conversation and the patient decided to have no further conversations (Catania center).

negatively impacted the experience of both the PwPMS and her SO. Two main challenges emerged: one was the uncertainty of actually implementing one's choices, and the other was the difficulty of SOs in fully understanding the commitment required by their role as a trustee.

Table 4. Exemplifying quotations from the interviews.

| Theme | Subtheme | Quotation(s) |
|---|---|---|
| **Existential significance** | **Relevance** | "I was... happy. Because I was able to write down on paper what my future will be, in short. I would say [the booklet was] very clear, and exhaustive: looking at what is written there, a person understands the will of the patient, in short. Even if I will be no longer able to make decisions, all my decisions are there" [126-PwPMS] "It means to be able 'to put a full stop' and not just that it was a nice decision on a cloud" [209-PwPMS] "More than important it's… 'heavy', in the sense that it's still... something that's... not easy" [401-SO] |
| | **Partnership** | "Sometimes a person risks choosing a healthcare path and wasting time on it. Competent people could help here because it's their job. Like the doctor who is guiding us, or something like that" [313-PwPMS] "[Doctor] made me feel comfortable because she asked me questions... and left me talk without... without asking, the doctor managed to talk to me, and I tried to explain all my problems" [508-PwPMS] "I think it is very important that the person knows what we are talking about. In the sense that he/she knows what it means to live with multiple sclerosis, how far one can go. In my opinion this is important, because often a person struggle to realize it, from the outside. Also, because there are patients who have... such mild symptoms all their lives that they don't provoke any problems. And so, logically, you don't think about an end of life" [401-SO] |
| **Impact** | **Emotional costs** | "A bit of a 'bitter taste in the mouth', that is. A feeling of......facing the end of life. That sometimes you think you're immortal, you never think about it" [126-PwPMS] "It's difficult, however, to approach a discussion like that. It must always be done in a certain way, with extreme delicacy because, as I said, he's young, so it's still a bit scary to think about something like that. Honestly, no matter how much one gets used to certain topics, it's always a bit frightening, that's what I mean (…). On the one hand it's scary and on the other hand it's also a relief because you share with other people these problems that a person has every day anyway. From a personal point of view, I think it also helps the relatives" [508-SO] |
| | **On family relationships** | "In case I... maybe have problems. We also put the children as trustee. We have three children and so... we've somehow involved them too, in case maybe I have problems. Maybe that also gives me some strength" [401-SO] "There is only one issue I would like to change – yes? the place where I will spend the end of my life, I had said a hospice, instead I would like at home…<br>▪ Did you reconsider that after a few days? Or recently…<br>Yes, talking about it within the family" [126-PwPMS] |
| **Preparedness** | **Reducing anxiety** | "Look, now honestly with the fact that my husband had already been 'fighting' for it [ACP] for some time, he talked about it, anyway he tried somehow everything to see if there was a possibility of having this opportunity. So, when they proposed it [ACP] to us, let's say my husband was happy, so, consequently, I was already prepared for it. (…) At the proposal, [I did not feel] anguish, no. No, because living with my husband, I understand that for [such patients] it is certainly their right" [401-SO] |
| | **Increasing difficulties** | "No, thinking back to the [ACP] interview, I say: It was hard…<br>(chuckle) I never thought. I never thought that this moment would come, that's it" [126-PwPMS]<br>▪ "It was not the right time for your sister, in your opinion?" [interviewer]<br>"Not at that time (…). She was not thinking at that time."<br>▪ "So, it was just a wrong moment, right?" [interviewer]<br>"From my point of view, yes. (…) During [the interview] honestly, I saw Dr. X a lot in difficulty (…) because perhaps she hadn't yet grasped that my sister hadn't understood the purpose of this thing" [502-SO] |
| **Challenges** | | |
| | **Practicability** | "Can my choices now be put into practice?" [126-PwPMS] |
| | **Commitment** | |
| | | ▪ "Have you been identified by your brother as a trustee?" [interviewer]<br>"I don't remember it formally, but it is very likely, due to the lack of credible alternatives, let's say (...). I don't remember, but it is quite inevitable that I was appointed by my brother" [507-SO] |

PwPMS, person with progressive multiple sclerosis; SO, significant other.

We held two focus groups with 14 clinicians from the six participating centers (clinicians' characteristics are not given to prevent their identification). Focus group guide is reported in S3 File. Fig 2 summarizes the facilitators and barriers identified using NPT constructs. *Coherence* was facilitated by clear study objectives and well-defined roles within the ACP process. Barriers included neurologists feeling pressured by PwPMS or SOs to skip thorough discussions, and a tendency to start ACP discussion in the late phase of the disease trajectory.

| FACILITATORS | BARRIERS |
|---|---|
| **Coherence**<br>*Understanding the intervention's purpose, components and potential benefits, both at an individual and collective level* | |
| - Clear study objectives and well-defined roles (*Differentiation/Individual specification*)<br>- ACP conversations aligned with routine practice, allowing PwPMS to articulate their preferences (*Internalization*) | - Pressure of PwPMS or SOs to skip booklet's contents<br>- ACP discussions typically delayed until clinical conditions worsen (*Internalization*) |
| **Cognitive Participation**<br>*The commitment and engagement of individuals and groups in the implementation process, including their willingness to adopt and support the new practice* | |
| - Study posters and flyers set the agenda and piqued PwPMS curiosity (*Enrollment*)<br>- Psychological support crucial for assessing PwPMS readiness for ACP (*Enrollment*)<br>- Attempts to convey to colleagues what clinicians have learned during training (*Initiation*) | - Lack of legitimization of ACP practice at MS centers (*Initiation*)<br>- Lack of psychological support (*Enrollment*)<br>- Resistance from colleagues and PwPMS unpreparedness<br>- Difficulties in sharing training insights within the teams |
| **Collective Action**<br>*The operational work that people do to enact a set of practices* | |
| - Interprofessional collaboration during ACP conversations (*Interactional workability*)<br>- All the MS team trained in ACP and in dealing with PwPMS/SOs emotions and worries (*Skill set workability*)<br>- Personalized, non-mechanical use of the booklet | - Rigid time schedule and use of the booklet<br>- Practical difficulty with space and time for ACP provision (*Interactional workability*)<br>- Lack of integration of ACP practice, and of a network of ACP-trained clinicians (*Contextual integration*) |
| **Reflexive Monitoring**<br>*The appraisal work that people do to assess and understand the ways that a new set of practices affect them and others around them* | |
| - In addition to help clinicians in supporting PwPMS EOL care choices, the training programme improved their awareness and ability to discuss psychosocial and existential issues<br>- The booklet is a useful guide despite some limitations<br>- Positive, enriching experience changing clinicians' attitude towards ACP | - Applying the training in current clinical practice is a challenge (*Systematization*)<br>- Misalignment between training expectations and recruitment outcomes<br>- Uncertainty about ACP promotion because of topic complexity (*Individual appraisal*)<br>- Unclear distinction between AD and ACP (*Reconfiguration*) |

*NORMALIZATION PROCESS THEORY*

**Fig 2. Facilitators and barriers identified in our analysis of focus groups with clinicians using the Normalization Process Theory constructs.**

*Cognitive participation* was facilitated by promoting ACP through study posters; moreover, clinicians considered the availability of psychological support to assess PwPMS readiness to ACP a resource. Barriers included the fact that ACP practice is rare, the unpreparedness of PwPMS in discussing disease worsening and EOL issues, and the difficulty in

sharing the training insights within the team. Clinicians suggested more comprehensive team training to enhance ACP's legitimacy and effectiveness.

*Collective action* benefited from collaboration among professionals with diverse competences. Clinicians emphasized the need for thorough ACP training and the primacy of the relationship over use of the booklet. To minimize barriers, particularly time and space availability, clinicians proposed reorganizing activities, embedding ACP in MS care pathways, and establishing a network of ACP-trained professionals.

*Reflexive monitoring* showed that clinicians described their participation in the trial as a positive, enriching experience, which changed their perspective towards ACP, increased their relational capabilities, and their openness in dealing with goals of care discussions. However, clinicians had difficulty in actively promoting ACP in their work context due to the atypia of the topic (*individual appraisal*). The booklet was helpful despite some limitations (redundancy and hastily presentation of EOL choices). Another barrier was the clinician's difficulty in distinguishing between advance directives/living wills (completed by any adult person with decisional capacity) and ACP. Finally, a trial-specific barrier was the impractical trial platform (Box 1).

Participants made suggestions to improve the intervention, mainly by letting the neurologist organize the time of ACP discussion, using the booklet flexibly (as a tool to be completed or a starting point for reflection), making ACP traceable; including other professionals (e.g., palliative care professionals) in the multidisciplinary team; identifying trigger points to propose ACP (e.g., transition to a progressive MS phase, unplanned hospitalizations); have dedicated space and time for ACP conversation; maintaining ACP training within the MS team and integrating it with case-based discussion.

The NoMAD subscale scores ranged between 79.6 (*collective action*) and 91.1 (*coherence*) out of a maximum value of 100 (S4 Table).

---

**Box 1. Findings of each pre-specified objective [9] of the study. ACP-Doc, advance care plan document; MSQOL-29, 29 item Multiple Sclerosis Quality of Life; PwPMS, person with progressive multiple sclerosis; SO, significant other; ZBI, Zarit Burden Interview.**

| Objective | Finding |
|---|---|
| How many PwPMS accepted the invitation to participate in the study | 19/75 (25%) |
| How many PwPMS receive the intervention | 19/19 (100%) |
| Recruitment and refusal rates, and 6-month follow-up rates | • 1 PwPMS refused to be recorded during the first ACP conversation and to complete all PROMs<br>• 1 PwPMS did not complete all PROMs at 6-month follow-up<br>• 2 SOs did not complete ZBI at baseline<br>• 2 SOs refused to complete ZBI at 6-month follow-up |
| ACP-Doc completion during the 6-month follow-up (primary study outcome). | 11/19 (58%) |
| Serious adverse events and adverse events during the 6-month follow-up | • 1 Serious adverse event, unrelated to the intervention<br>• 25 Adverse events, 3/25 possibly related to the intervention |
| Changes in the secondary outcome measures | • Significant increase in anxiety and worsening in MSQOL-29 Mental Health Composite after ACP conversation and at 6-month follow-up<br>• No change in the other outcome measures<br>• These findings should be interpreted with caution (see Discussion, 'Study limitations' section) |
| Acceptability of the recruitment processes, assessments, intervention delivery and secondary outcome measures (qualitative sub-study) | • Acceptable to clinicians, but widespread involvement of the team is vital<br>• Concurrent clinicians' commitments as a challenge<br>• Acceptable to PwPMS and SOs, but some difficulties in using the trial platform |
| • Barriers and facilitators to implementing ACP in PwPMS,<br>• and the influence of the clinical setting | • See Fig 2<br>• Inpatient rehab setting allows sufficient time and space for conversations, while outpatient setting allows more flexible follow-up conversation scheduling |
| Sample size estimation for a subsequent phase III trial, should this be feasible | Not applicable (need to enrich the intervention with a component targeting PwPMS and SOs) |

## Discussion

This study examined the feasibility of an intervention to improve implementation of ACP for PwPMS using a mixed-methods approach. Study findings suggest that the training programme and use of the booklet helped the clinicians in guiding PwPMS reflection on their personal values and preferences, treatment options and their consequences, and devise an ACP-Doc. Trial procedures and outcome measure collection are also feasible except for a difficult-to-use trial platform by PwPMS and SOs, including poor visibility of the CPS score validation button (Box 1). The main study challenge was that, even if we extended the recruitment period from six to 12 months, and placed study posters in the waiting rooms, the planned sample of 40 PwPMS was not reached due to two-thirds of eligible PwPMS refusing participation (Box 1). This high PwPMS refusals point to the need for a broader cultural change towards EOL issues and discussion (reconfiguration). For this reason, we decided that moving to the next project phase is premature, and a new intervention component targeting PwPMS and SOs is required. This is in line with the suggestion of a multiple-component ACP intervention targeting clinicians and patients simultaneously [29], measuring first, the level of readiness for ACP of patients and carers [30].

Given that this was a small pilot/feasibility study, the sample size was not powered to detect changes in the proposed outcome measures of the definitive (cluster) randomized controlled trial (RCT). However, it is worth mentioning the increase in PwPMS anxiety symptoms after the first ACP conversation and at six-month follow-up. As from a systematic review on ACP interventions [31], only two RCTs assessed patient anxiety symptoms, both in advanced cancer sufferers: Bernacki et al. found significant reduction in the proportion of patients with moderate-to-severe anxiety approximately 12 weeks post-intervention [32]; Clayton et al. did not find any between-group difference in mean State Anxiety Inventory score change three weeks post-intervention [33]. We believe that the existential issues covered during the ACP discussion may lead to an increase of anxiety, and this response should not be necessarily considered as the expression of a psychopathological disorder. Nevertheless, it is important that this symptom is properly identified and managed by the caring physician. A deeper understanding of the main determinants of this psychological reaction deserves further investigation.

Two main barriers for ACP discussions in MS were identified in a literature review: the long and uncertain MS trajectory, and lack of ACP communication skills and confidence of clinicians [34]. The MS trajectory challenges the inclusion of long-term outcomes (chiefly concordance between preferred and received EOL care and treatments [35]) in the typical timeframe of a clinical trial. For this reason and in coherence with the ACP principles, we agreed not to narrow the inclusion criteria only to PwPMS in the late disease stage. Nevertheless, one enrolled PwPMS had a SAE (Table 3) and on that occasion the intensive care clinicians applied his ACP-Doc. Concerning clinicians' skills, a recent online survey revealed that only 18% of Italian residents in neurology had received training in ACP and only 13% had participated in the ACP process, half within their residency programme [36]. Consistent with this survey, none of the clinicians who participated in the present study had previous experience of ACP. We explored the QOC scores of one neurologist who participated in 5/19 ACP discussions, and found an increase in QOC-Doc scores over time, with values ranging between 25 (first conversation) and 80 (fourth and fifth conversation) out of a maximum value of 100 (Spearman's rho 0.97, $p = 0.005$). Moreover, structured observation of the audio-recorded ACP conversations (manuscript under review) revealed that in the first conversation the neurologist was quite anchored to the booklet, while subsequently she used the tool in a less mechanistic way.

## Study limitations

Although valuable for assessing the feasibility and potential of a larger study, pilot and feasibility studies are typically underpowered. Additionally, we did not reach the target number of participants and used (as from the study protocol) advanced statistical methods (i.e., mixed-method analysis and generalized estimating equations). Finally, we assessed

several secondary outcome measures, without adjusting for multiplicity testing. Therefore, the results of the quantitative analysis should be interpreted with caution.

## Conclusions

The ConCure-SM intervention is feasible and acceptable to PwPMS, their SOs, and clinicians. Our findings suggest that the training programme and use of the booklet have the potential to support PwPMS reflection on their personal values and preferences, clarification of treatment options and their consequences, and devise of an ACP-Doc. However, before moving on to the evaluation phase it is necessary to enrich the intervention with a component that improves the ACP preparedness of both PwPMS and their SOs. In addition, flexibility in intervention delivery across different contexts (outpatient versus inpatient services) should be pursued, yet preserving the integrity of its core components.

## Supporting information

**S1 File. CONSORT 2010 extension to randomized pilot and feasibility trials checklist.**
(PDF)

**S2 File. ConCure-SM study protocol.**
(PDF)

**S3 File. Interview/focus group guides.**
(PDF)

**S4 File. COREQ checklist.**
(PDF)

**S1 Box. Conditions that would make advance care planning (ACP) relevant.**
(PDF)

**S1 Table. Characteristics of subjects at screening across centers.**
(PDF)

**S2 Table. Participant's characteristics at baseline across centers.**
(PDF)

**S3 Table. Per protocol analysis of the secondary outcome measures.**
(PDF)

**S4 Table. NoMAD frequencies of agreement and total scores of 13 clinicians (7 neurologists, 6 other professionals) from the six participating centers.**
(PDF)

**S1 Fig. Scatter plots of Hospital Anxiety and Depression Scale (HADS)-Anxiety scores at baseline, after the first advance care planning (ACP) conversation (T1), and at 6-month follow-up (T2) (per-protocol data).** Codes of people with progressive multiple sclerosis who completed the ACP-Document are reported in bold.
(PDF)

**S2 Fig. Scatter plots of Hospital Anxiety and Depression Scale (HADS)-Depression scores at baseline, after the first advance care planning (ACP) conversation (T1), and at 6-month follow-up (T2) (per-protocol data).** Codes of people with progressive multiple sclerosis who completed the ACP-Document are reported in bold.
(PDF)

**S3 Fig. Scatter plots of 4-item ACP Engagement survey scores at baseline, after the first advance care planning (ACP) conversation (T1), and at 6-month follow-up (T2) (per-protocol data).** Codes of people with progressive multiple sclerosis who completed the ACP-Document are reported in bold.
(PDF)

**S4 Fig. Box plots of MSQOL-29 Physical Health Composite (A) and MSQOL-29 Mental Health Composite scores (B) at baseline and at 6-month follow up (T2).** The boxes represent the interquartile range, horizontal lines inside boxes represent medians and tails represent the 5th–25th and 75th–95th percentile range. Dots are outliers. MSQOL-29, 29-item Multiple Sclerosis Quality of Life; MHC, mental health composite; PHC, physical health composite.
(PDF)

**S5 Fig. Box plots of ZBI scores at baseline, after the first advance care planning conversation (T1), and at 6-month follow up (T2).** The boxes represent the interquartile range, horizontal lines inside boxes represent medians and tails represent the 5th–25th and 75th–95th percentile range. Dots are outliers. ZBI, Zarit Burden Interview.
(PDF)

## Acknowledgments

We thank all of the PwPMS and SOs who participated in the study.

**ConCure-SM project investigators**

**ConCure-SM Steering Committee:** LDP, SV, Michela Bruzzone, Italian Multiple Sclerosis Society, Genoa, Italy; AG, MGG, PK, AL, SM, FP, EP, CS, AGi, AS.

**Data Safety and Monitoring Committee:** David Oliver, The Tizard Centre, University of Kent, Canterbury, UK (Chair); Kevin Brazil, School of Nursing and Midwifery, Queen's University of Belfast, Belfast, Northern Ireland, UK; Bobbie Farsides, Brighton & Sussex Medical School, Falmer, Brighton, United Kingdom; Luciano Orsi, The Italian Society of Palliative Care (SICP), Milan, Italy; Carlo Peruselli, SICP, Milan, Italy.

**Data Management and Analysis Committee:** AGi, MF.

**Qualitative Analysis Panel:** LDP, SV, MP, LG, KM, RMZ.

**HP Training Panel:** LDP, SV, Marta Cascioli, Hospice La Torre sul Colle, Spoleto, Italy; KM, EP, MR, AS.

**Linguistic Validation Panel:** MF, PK, SV, AGi, AS.

**Enrolling Centers and Investigators:** Department of Neuroscience, Biomedicine and Movement Sciences, University of Verona; Unit of Neurology, Borgo Roma Hospital, Azienda Ospedaliera Universitaria Integrata Verona, Verona: AG, Angelica Filosa, Riccardo Orlandi. Department of Rehabilitation M. L. Novarese Hospital, Moncrivello, Vercelli: CS, Erica Grange. Multiple Sclerosis Center, Azienda USL-IRCCS di Reggio Emilia, Reggio Emilia: SM, Francesca Sireci. UOSI Riabilitazione Sclerosi Multipla, IRCCS Istituto delle Scienze Neurologiche di Bologna; Dipartimento di Scienze Biomediche e Neuromotorie, Università di Bologna, Bologna: AL, Federica Pinardi, Loredana Sabbatini, Cinzia Scandellari, Elisa Ferriani. Fondazione IRCCS Santa Lucia, Rome: MGG, Giorgia Presicce, Luigi Iasevoli. University Hospital Policlinico Vittorio Emanuele, Catania: FP, Clara Grazia Chisari, Simona Toscano.

## Author contributions

**Conceptualization:** Alessandra Solari, Ludovica De Panfilis, Simone Veronese.

**Data curation:** Roberta Martina Zagarella, Luca Ghirotto, Mariangela Farinotti, Andrea Giordano.

**Formal analysis:** Ludovica De Panfilis, Roberta Martina Zagarella, Marta Perin, Andrea Giordano, Simone Veronese.

**Funding acquisition:** Alessandra Solari.

**Investigation:** Alberto Gajofatto, Maria Grazia Grasso, Alessandra Lugaresi, Sara Montepietra, Francesco Patti, Claudio Solaro.

**Methodology:** Alessandra Solari, Luca Ghirotto.

**Project administration:** Mariangela Farinotti.

**Supervision:** Alessandra Solari, Katia Mattarozzi, Eugenio Pucci, Michela Rimondini, Simone Veronese.

**Writing – original draft:** Alessandra Solari, Luca Ghirotto, Andrea Giordano.

**Writing – review & editing:** Alessandra Solari, Ludovica De Panfilis, Roberta Martina Zagarella, Luca Ghirotto, Mariangela Farinotti, Alberto Gajofatto, Maria Grazia Grasso, Paola Kruger, Alessandra Lugaresi, Katia Mattarozzi, Sara Montepietra, Francesco Patti, Eugenio Pucci, Michela Rimondini, Claudio Solaro, Marta Perin, Andrea Giordano, Simone Veronese.

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
