## [Decision Letter · Decision Letter 0]

25 Apr 2025

Dear Dr. Solari,

Thank you for submitting your manuscript to PLOS ONE. After careful consideration, we feel that it has merit but does not fully meet PLOS ONE’s publication criteria as it currently stands. Therefore, we invite you to submit a revised version of the manuscript that addresses the points raised during the review process.

We look forward to receiving your revised manuscript.

Kind regards,

Nayanatara Arun Kumar

Academic Editor

PLOS ONE

**Journal Requirements:**

1. When submitting your revision, we need you to address these additional requirements. Please ensure that your manuscript meets PLOS ONE's style requirements, including those for file naming. The PLOS ONE style templates can be found at https://journals.plos.org/plosone/s/file?id=wjVg/PLOSOne_formatting_sample_main_body.pdf and https://journals.plos.org/plosone/s/file?id=ba62/PLOSOne_formatting_sample_title_authors_affiliations.pdf 2. Thank you for stating in your Funding Statement: The study was supported by FISM - Fondazione Italiana Sclerosi Multipla – cod. 2020/R-Multi/024 and financed or co-financed with the ‘5 per mille’ public funding to AS. The funding source had no role in study design, data collection, data analysis, data interpretation or report writing. We also thank the Italian Ministry of Health (RRC).  Please provide an amended statement that declares *all* the funding or sources of support (whether external or internal to your organization) received during this study, as detailed online in our guide for authors at http://journals.plos.org/plosone/s/submit-now.  Please also include the statement “There was no additional external funding received for this study.” in your updated Funding Statement. Please include your amended Funding Statement within your cover letter. We will change the online submission form on your behalf. 3. Thank you for stating the following in the Competing Interests section: AL reports grants from Novartis, during the conduct of the study; personal fees from Biogen, Merck Serono, Mylan, Novartis, Roche, Sanofi/Genzyme, Teva and FISM. FP received personal compensation for serving on advisory board and/or speaking activities by Almirall, Bayer, Biogen, Bristol Meyers & Squibb, Merck, Novartis Roche, Sanofi and TEVA; he further received research grants by Biogen Italy, Biogen Global, Merck, University of Catania, FISM and Reload Onlus Patients Association. AS received personal compensation for serving on advisory board and speaking activities by Almirall and Merck. All the other authors have no competing interests. We note that one or more of the authors are employed by a commercial company.  a. Please provide an amended Funding Statement declaring this commercial affiliation, as well as a statement regarding the Role of Funders in your study. If the funding organization did not play a role in the study design, data collection and analysis, decision to publish, or preparation of the manuscript and only provided financial support in the form of authors' salaries and/or research materials, please review your statements relating to the author contributions, and ensure you have specifically and accurately indicated the role(s) that these authors had in your study. You can update author roles in the Author Contributions section of the online submission form. Please also include the following statement within your amended Funding Statement. “The funder provided support in the form of salaries for authors, but did not have any additional role in the study design, data collection and analysis, decision to publish, or preparation of the manuscript. The specific roles of these authors are articulated in the ‘author contributions’ section.”If your commercial affiliation did play a role in your study, please state and explain this role within your updated Funding Statement.  b. Please also provide an updated Competing Interests Statement declaring this commercial affiliation along with any other relevant declarations relating to employment, consultancy, patents, products in development, or marketed products, etc.   Within your Competing Interests Statement, please confirm that this commercial affiliation does not alter your adherence to all PLOS ONE policies on sharing data and materials by including the following statement: "This does not alter our adherence to  PLOS ONE policies on sharing data and materials.” (as detailed online in our guide for authors http://journals.plos.org/plosone/s/competing-interests) . If this adherence statement is not accurate and  there are restrictions on sharing of data and/or materials, please state these. Please note that we cannot proceed with consideration of your article until this information has been declared. Please include both an updated Funding Statement and Competing Interests Statement in your cover letter. We will change the online submission form on your behalf. 4. Please note that your Data Availability Statement is currently missing the repository name. If your manuscript is accepted for publication, you will be asked to provide these details on a very short timeline. We therefore suggest that you provide this information now, though we will not hold up the peer review process if you are unable. 5. PLOS requires an ORCID iD for the corresponding author in Editorial Manager on papers submitted after December 6th, 2016. Please ensure that you have an ORCID iD and that it is validated in Editorial Manager. To do this, go to ‘Update my Information’ (in the upper left-hand corner of the main menu), and click on the Fetch/Validate link next to the ORCID field. This will take you to the ORCID site and allow you to create a new iD or authenticate a pre-existing iD in Editorial Manager. 6. We note that you have included the phrase “data not shown” in your manuscript. Unfortunately, this does not meet our data sharing requirements. PLOS does not permit references to inaccessible data. We require that authors provide all relevant data within the paper, Supporting Information files, or in an acceptable, public repository. Please add a citation to support this phrase or upload the data that corresponds with these findings to a stable repository (such as Figshare or Dryad) and provide and URLs, DOIs, or accession numbers that may be used to access these data. Or, if the data are not a core part of the research being presented in your study, we ask that you remove the phrase that refers to these data. 7. We note that there is identifying data in the Supporting Information files “S3 and S4 Tables.” Due to the inclusion of these potentially identifying data, we have removed this file from your file inventory. Prior to sharing human research participant data, authors should consult with an ethics committee to ensure data are shared in accordance with participant consent and all applicable local laws. Data sharing should never compromise participant privacy. It is therefore not appropriate to publicly share personally identifiable data on human research participants. The following are examples of data that should not be shared: -Name, initials, physical address-Ages more specific than whole numbers-Internet protocol (IP) address-Specific dates (birth dates, death dates, examination dates, etc.)-Contact information such as phone number or email address-Location data-ID numbers that seem specific (long numbers, include initials, titled “Hospital ID”) rather than random (small numbers in numerical order) Data that are not directly identifying may also be inappropriate to share, as in combination they can become identifying. For example, data collected from a small group of participants, vulnerable populations, or private groups should not be shared if they involve indirect identifiers (such as sex, ethnicity, location, etc.) that may risk the identification of study participants. Additional guidance on preparing raw data for publication can be found in our Data Policy (https://journals.plos.org/plosone/s/data-availability#loc-human-research-participant-data-and-other-sensitive-data) and in the following article: http://www.bmj.com/content/340/bmj.c181.long. Please remove or anonymize all personal information (<specific identifying information in file to be removed>), ensure that the data shared are in accordance with participant consent, and re-upload a fully anonymized data set. Please note that spreadsheet columns with personal information must be removed and not hidden as all hidden columns will appear in the published file.

**Additional Editor Comments:**

Dear Authors

Please go through the queries asked by the reviewer and do the corrections accordingly and resubmit for the editorial decision, All the best

With regards and Best Wishes

Dr. Nayanatara

Reviewers' comments:

Reviewer's Responses to Questions

**Comments to the Author**

1. Is the manuscript technically sound, and do the data support the conclusions?

Reviewer #1: Yes

2. Has the statistical analysis been performed appropriately and rigorously?

Reviewer #1: Yes

3. Have the authors made all data underlying the findings in their manuscript fully available?

Reviewer #1: Yes

4. Is the manuscript presented in an intelligible fashion and written in standard English?

Reviewer #1: Yes

**Reviewer #1:**  The authors presented a pilot and feasibility study of advance care planning in Multiple sclerosis. The study was well planned, advanced statistical methods were used to analyse the data based on mixed method, but the sample size was not large enough.

Being a pilot and feasibility study, small sample size can be acceptable. However, large sample statistics were used to analyse small sample and hence quantitative part of the analysis is not reliable. I have few other comments to make

1. The Letter “P” in P-values should be italic throughout

2. At the end of the methods section in the Abstract need to mention that a mixed method analysis was done.

3. So many outcome measures scales were used based on a small sample size, yet adjustment in p-values were not done for multiplicity. This is a big limitation needs to be corrected or mentioned as a limitation.

4. As the study could not recruit the desired number of patients, inferential statistics (p-values) based on quantitative results are not reliable as they are also affected by the multiple testing.

5. In line 203, it says that generalized estimating equations were used for secondary outcomes. GEE is also a large sample procedure and hence would not give reliable results.

6. In line 205, Section Statistics, says that multiple imputation of missing values was done. A bit more information needs to be presented to show the extent of the missing values.

7. Qualitative analyses are good.

8. As this is a pilot and feasibility study, all the limitations for quantitative analysis need to be mentioned in appropriate places.

**Do you want your identity to be public for this peer review?** For information about this choice, including consent withdrawal, please see our Privacy Policy

Reviewer #1: **Yes: ** Dr Shah-Jalal Sarker

---

## [Author Response · Author response to Decision Letter 1]

30 Apr 2025

RESPONSE TO REVIEWERS

1. The authors presented a pilot and feasibility study of advance care planning in Multiple sclerosis. The study was well planned, advanced statistical methods were used to analyse the data based on mixed method, but the sample size was not large enough. Being a pilot and feasibility study, small sample size can be acceptable. However, large sample statistics were used to analyse small sample and hence quantitative part of the analysis is not reliable.

R: We agree with the reviewer that the quantitative analysis findings should be interpreted with caution. We have mentioned this in the Discussion, as follows: “Study limitations. Although useful for assessing the feasibility and potential of a larger study, pilot and feasibility studies are typically underpowered. Additionally, we did not reach the target number of participants and used (as from the study protocol) advanced statistical methods (i.e., mixed-method analysis and generalized estimating equations). Finally, we assessed several secondary outcome measures, without adjusting for multiplicity testing. Therefore, the results of the quantitative analysis should be interpreted with caution.”

2. The Letter “P” in P-values should be italic throughout.

R: We have made changes following reviewer’s suggestion.

3. At the end of the methods section in the Abstract need to mention that a mixed method analysis was done.

R: We have revised the Abstract as suggested.

4. So many outcome measures scales were used based on a small sample size, yet adjustment in p-values were not done for multiplicity. This is a big limitation needs to be corrected or mentioned as a limitation.

R: See our response to point 1 above.

5. As the study could not recruit the desired number of patients, inferential statistics (p-values) based on quantitative results are not reliable as they are also affected by the multiple testing.

R: See our response to point 1 above.

6. In line 203, it says that generalized estimating equations were used for secondary outcomes. GEE is also a large sample procedure and hence would not give reliable results.

R: See our response to point 1 above.

7. In line 205, Section Statistics, says that multiple imputation of missing values was done. A bit more information needs to be presented to show the extent of the missing values.

R: This information is provided in the Results, section “Adverse Events and Attrition” (lines 261-269).

8. As this is a pilot and feasibility study, all the limitations for quantitative analysis need to be mentioned in appropriate places.

R: In addition to the Discussion (see our response to point 1 above) we have mentioned these limitations in the Results (section “Other outcome measures”, lines 93-95), and in the Box.

---

## [Decision Letter · Decision Letter 1]

15 Jul 2025

Dear Dr. Solari,

Please do the needful, please address each comments 

Please submit your revised manuscript by Aug 29 2025 11:59PM. If you will need more time than this to complete your revisions, please reply to this message or contact the journal office at plosone@plos.org . A rebuttal letter that responds to each point raised by the academic editor and reviewer(s). You should upload this letter as a separate file labeled 'Response to Reviewers'.A marked-up copy of your manuscript that highlights changes made to the original version. You should upload this as a separate file labeled 'Revised Manuscript with Track Changes'.An unmarked version of your revised paper without tracked changes. You should upload this as a separate file labeled 'Manuscript'.

We look forward to receiving your revised manuscript.

Kind regards,

Nayanatara Arun Kumar

Academic Editor

PLOS ONE

Journal Requirements:

Additional Editor Comments:

Dear Authors

I have gone through the revision file as suggested by the respectable reviewers , based on this the manuscript is accepted for publication . Congratulations to all the authors

Reviewers' comments:

Reviewer's Responses to Questions

**Comments to the Author**

Reviewer #1: (No Response)

2. Is the manuscript technically sound, and do the data support the conclusions?

Reviewer #1: Yes

3. Has the statistical analysis been performed appropriately and rigorously?

Reviewer #1: Yes

4. Have the authors made all data underlying the findings in their manuscript fully available?

Reviewer #1: Yes

5. Is the manuscript presented in an intelligible fashion and written in standard English?

Reviewer #1: Yes

Reviewer #1: In the section of other outcome measures in the Results: These findings should be interpreted with caution (see Discussion, ‘Study limitations’ section). Please add, "due to small sample size and multiplicity problem".

**Do you want your identity to be public for this peer review?** For information about this choice, including consent withdrawal, please see our Privacy Policy

Reviewer #1: **Yes: ** Dr Shah-Jalal Sarker

---

## [Author Response · Author response to Decision Letter 2]

16 Jul 2025

RESPONSE TO REVIEWER #1

In the section of other outcome measures in the Results: These findings should be interpreted with caution (see Discussion, ‘Study limitations’ section). Please add, "due to small sample size and multiplicity problem".

R: We have added the phrase, as suggested.

---

## [Decision Letter · Decision Letter 2]

13 Aug 2025

Advance care planning in multiple sclerosis (ConCure-SM): A multicenter single-arm pilot and feasibility study

PONE-D-25-03850R2

Dear Dr. Solari

We’re pleased to inform you that your manuscript has been judged scientifically suitable for publication and will be formally accepted for publication once it meets all outstanding technical requirements.

Kind regards,

Nayanatara Arun Kumar

Academic Editor

PLOS ONE

Additional Editor Comments (optional):

Dear Authors

Congratulations to you and move forward with more in depth research

regards

Dr, Nayanatara Arun kumar

Reviewers' comments:

Reviewer's Responses to Questions

**Comments to the Author**

Reviewer #1: All comments have been addressed

2. Is the manuscript technically sound, and do the data support the conclusions?

Reviewer #1: Yes

3. Has the statistical analysis been performed appropriately and rigorously?

Reviewer #1: Yes

4. Have the authors made all data underlying the findings in their manuscript fully available?

Reviewer #1: Yes

5. Is the manuscript presented in an intelligible fashion and written in standard English?

Reviewer #1: Yes

Reviewer #1: (No Response)

**Do you want your identity to be public for this peer review?** For information about this choice, including consent withdrawal, please see our Privacy Policy

Reviewer #1: **Yes: ** Dr Shah-Jalal Sarker

---

## [Editor Report · Acceptance letter]

PONE-D-25-03850R2

PLOS ONE

Dear Dr. Solari,

I'm pleased to inform you that your manuscript has been deemed suitable for publication in PLOS ONE. Congratulations! Your manuscript is now being handed over to our production team.

Kind regards,

on behalf of

Dr. Nayanatara Arun Kumar

Academic Editor

PLOS ONE